# Whole-organism eQTL mapping at cellular resolution with single-cell sequencing

Eyal Ben-David[1,2]*, James Boocock[1], Longhua Guo[1], Stefan Zdraljevic[1], Joshua S Bloom[1]*, Leonid Kruglyak[1]*

[1]Department of Human Genetics, Department of Biological Chemistry, and Howard Hughes Medical Institute, University of California, Los Angeles, Los Angeles, United States; [2]Department of Biochemistry and Molecular Biology, Institute for Medical Research Israel-Canada, The Hebrew University School of Medicine, Jerusalem, Israel

**Abstract** Genetic regulation of gene expression underlies variation in disease risk and other complex traits. The effect of expression quantitative trait loci (eQTLs) varies across cell types; however, the complexity of mammalian tissues makes studying cell-type eQTLs highly challenging. We developed a novel approach in the model nematode *Caenorhabditis elegans* that uses single-cell RNA sequencing to map eQTLs at cellular resolution in a single one-pot experiment. We mapped eQTLs across cell types in an extremely large population of genetically distinct *C. elegans* individuals. We found cell-type-specific *trans* eQTL hotspots that affect the expression of core pathways in the relevant cell types. Finally, we found single-cell-specific eQTL effects in the nervous system, including an eQTL with opposite effects in two individual neurons. Our results show that eQTL effects can be specific down to the level of single cells.

*For correspondence:
eyal.bendavid@mail.huji.ac.il (EB-D);
jbloom@mednet.ucla.edu (JSB);
LKruglyak@mednet.ucla.edu (LK)

**Competing interests:** The authors declare that no competing interests exist.

## Introduction

Gene expression differences have a strong genetic basis, and genome-wide studies have identified thousands of regions affecting gene expression, termed expression quantitative trait loci (eQTLs) (*Albert and Kruglyak, 2015*). eQTLs have been found to underlie genetic associations with complex traits and diseases (*Albert and Kruglyak, 2015*; *Gusev et al., 2014*; *Hormozdiari et al., 2018*), and genome-wide eQTL mapping holds great promise for uncovering the molecular underpinnings of phenotypic variation.

Studies in purified blood cell populations (*Raj et al., 2014*; *Fairfax et al., 2012*; *Ishigaki et al., 2017*) and computational analyses in human tissues (*Donovan et al., 2020*; *Kim-Hellmuth et al., 2020*) indicate that many eQTLs are cell-type specific. Studying eQTLs in relevant tissues and cell types is thus important for understanding their role in trait variation (*Yao et al., 2020*). However, major challenges remain for cell-type-specific eQTL mapping. First, studying multiple tissues or cell types is laborious and expensive, and has mostly been limited to large consortia (*GTEx Consortium, 2020*). Additionally, diseases can be associated with specific subsets of cells that may not be readily separable from the rest of the tissue, limiting the insight that can be gleaned from studying bulk tissue samples (*van der Wijst et al., 2020*).

The recently developed technology of single-cell RNA sequencing (scRNA-seq) has the potential to significantly mitigate both of the above issues. In scRNA-seq, gene expression is profiled at the level of individual cells, and different cell populations are identified based on their gene expression profiles. This approach allows simultaneous measurement of expression in multiple tissues and cell types in a 'one-pot' experiment, reducing the number of samples that need to be processed and profiled separately. Furthermore, distinct cell types can be interrogated directly, facilitating eQTL mapping in disease-relevant but rare cell types.

**eLife digest** DNA sequences that differ between individuals often change the activation pattern of genes. That is, they change how, when, or why genes switch on and off. We call such DNA sequences 'expression quantitative trait loci', or eQTLs for short. Many of these eQTLs affect biological traits, but their effects are not always easy to predict. In fact, these effects can change with time, with different stress levels, and even in different types of cells. This makes studying eQTLs challenging, especially in organisms with many different types of cells.

Standard methods of studying eQTLs involve separately measuring which genes switch on or off under every condition and in each cell. However, a technology called single cell sequencing makes it possible to profile many cells simultaneously, determining which genes are switched on or off in each one. Applying this technology to eQTL discovery could make a challenging problem solvable with a straightforward experiment.

To test this idea, Ben-David et al. worked with the nematode worm *Caenorhabditis elegans*, a frequently-used experimental animal which has a small number of cells with well-defined types. The experiment included tens of thousands of cells from tens of thousands of genetically distinct worms. Using single cell sequencing, Ben-David et al. were able to find eQTLs across all the different cell types in the worms. These included many eQTLs already identified using traditional approaches, confirming that the new method worked. To understand the effects of some of these eQTLs in more detail, Ben-David et al. took a closer look at the cells of the nervous system. This revealed that eQTL effects not only differ between cell types, but also between individual cells. In one example, an eQTL changed the expression of the same gene in opposite directions in two different nerve cells.

There is great interest in studying eQTLs because they provide a common mechanism by which changes in DNA can affect biological traits, including diseases. These experiments highlight the need to compare eQTLs in all conditions and tissues of interest, and the new technique provides a simpler way to do so. As single-cell technology matures and enables profiling of larger numbers of cells, it should become possible to analyze more complex organisms. This could one day include mammals.

The size and complexity of mammalian tissues has so far limited the ability to use scRNA-seq for eQTL mapping, and studies have focused on purified cell populations and cell lines (*Cuomo et al., 2020*; *van der Wijst et al., 2018*). To pilot a whole-organism approach, we therefore turned to the model nematode *Caenorhabditis elegans*. One of the mainstays of modern genetic research, *C. elegans* has an invariant cell lineage that leads to each individual having the same number and identity of cells (*Sulston and Horvitz, 1977*). Decades of research have uncovered the functions of many of those cells, as well as expression markers that uniquely identify them (*Hall and Altun, 2007*; *Hobert et al., 2016*). These features and resources make *C. elegans* exceptionally well-suited for using scRNA-seq to map eQTLs across cell types in the natural physiological context of a whole animal, down to cellular resolution.

## Results

### Cell-type-specific eQTL mapping in a single one-pot experiment

Genome-wide eQTL mapping involves acquiring genotypes and gene expression profiles for a genetically diverse cohort. We recently developed a method, *C. elegans* extreme quantitative trait locus (ceX-QTL) mapping, for genetic analysis of complex traits in extremely large populations of segregants (*Burga et al., 2019*). The method takes advantage of a mutation in the gene *fog-2* that forces the normally hermaphroditic *C. elegans* to reproduce via obligate outcrossing, allowing us to propagate a large crossing experiment for multiple generations.

Here, we build on ceX-QTL by combining it with scRNA-seq to carry out eQTL mapping at cellular resolution in a single one-pot experiment (*Figure 1A*). In this approach, a large heterogeneous pool of cells from thousands of genetically distinct individuals is profiled using scRNA-seq, cell types are inferred by clustering scRNA-seq profiles and studying known cell-type markers, and genotype

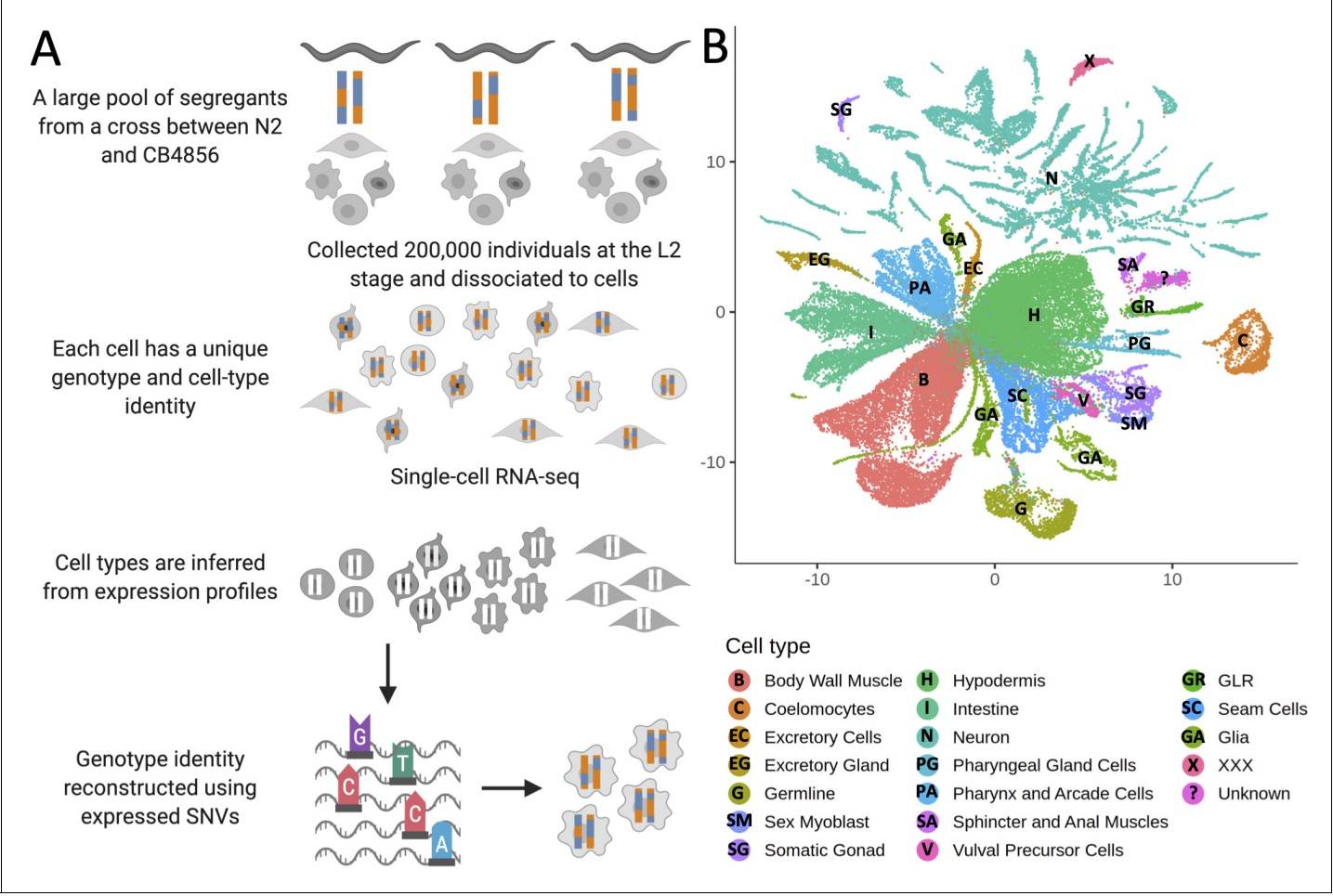

**Figure 1.** Whole-organism expression quantitative trait loci mapping with single-cell RNA-sequencing (scRNA-seq). (**A**) A large population of segregants is dissociated to single cells. Each cell in the suspension has an unknown genotype and cell-type identity. The suspension is profiled using scRNA-seq. Cell-type identity is inferred by clustering cells and comparing the expression of known marker genes. Genotypes are reconstructed from expressed single-nucleotide variants. (**B**) The Uniform Manifold Approximation and Projection of 55,508 scRNA-seq expression profiles from approximately 200,000 *C. elegans* F4 segregants collected at the L2 larval stage is shown. Cells are colored based on the inferred cell type. The online version of this article includes the following figure supplement(s) for figure 1:

**Figure supplement 1.** Observed representation of cell types in our dataset compared to expected.

information is reconstructed using expressed genetic variants, enabling eQTL mapping in multiple cell types simultaneously.

We propagated a cross between the laboratory strain N2 and a highly divergent isolate from Hawaii, CB4856, for four generations, generating a pool of 200,000 genetically distinct F4 segregants. We dissociated the segregant pool to single cells at the L2 larval stage and profiled the cells with scRNA-seq. We identified clusters in a Uniform Manifold Approximation and Projection (UMAP) of the dataset (*McInnes et al., 2018*; *Traag et al., 2019*; *Trapnell et al., 2014*) and determined their cell-type identities using known markers (*Cao et al., 2017*; *Packer et al., 2019*). Our final dataset comprises 55,508 cells classified into 19 different cell types (*Figure 1B*; *Supplementary file 1*—Table S1). The observed number of cells of each type was strongly correlated with the known cell-type abundance in L2 larvae (Spearman's ρ=0.87, p=$2.2\times10^{-6}$, *Figure 1—figure supplement 1*).

## De novo genotyping of scRNA-seq profiles using a hidden Markov model (HMM)

Most of the cells in our sample were expected to carry unique genotypes (Materials and methods, *Supplementary file 1*—Table S1). This design is advantageous for eQTL mapping because it maximizes the sample size (*Mandric et al., 2020*), but it requires de novo genotype calling because the genotype of each cell is unknown beforehand. Previous studies have shown that in addition to providing a readout of global gene expression, transcriptomic data can also be used for genotyping by leveraging variants in transcribed sequences (*Ronald et al., 2005*; *West et al., 2006*; *Kang et al., 2018*). To that end, we built a HMM for genotyping cells based on scRNA-seq data. The model calculates the posterior probability of the underlying genotypes for each individual based on three components: (1) prior probabilities for each of the possible genotypes, (2) emission probabilities for observing variant-informative reads given each of the possible genotypes, and (3) transition probabilities for recombination events occurring or not occurring between adjacent genotype-informative sites. Emission probabilities were calculated as previously described for low-coverage sequencing data under the assumption that the observed counts of reads for each possible allele at a genotype-informative site arise from a random binomial sampling of the alleles present and that sequencing errors occur independently (*Dodds et al., 2015*; *Bilton et al., 2018*). Transition probabilities at each variant position were derived from the genetic map we previously published for the N2 × CB4856 cross (*Rockman and Kruglyak, 2009*). Posterior probabilities of each genotype for each individual were calculated using the forward-backward algorithm (*Figure 2*; see Materials and methods for a full mathematical description of the model). To minimize loss of power due to genotyping errors, we used the posterior probabilities directly to map eQTLs in a negative binomial modeling framework instead of assigning deterministic genotype calls (Materials and methods).

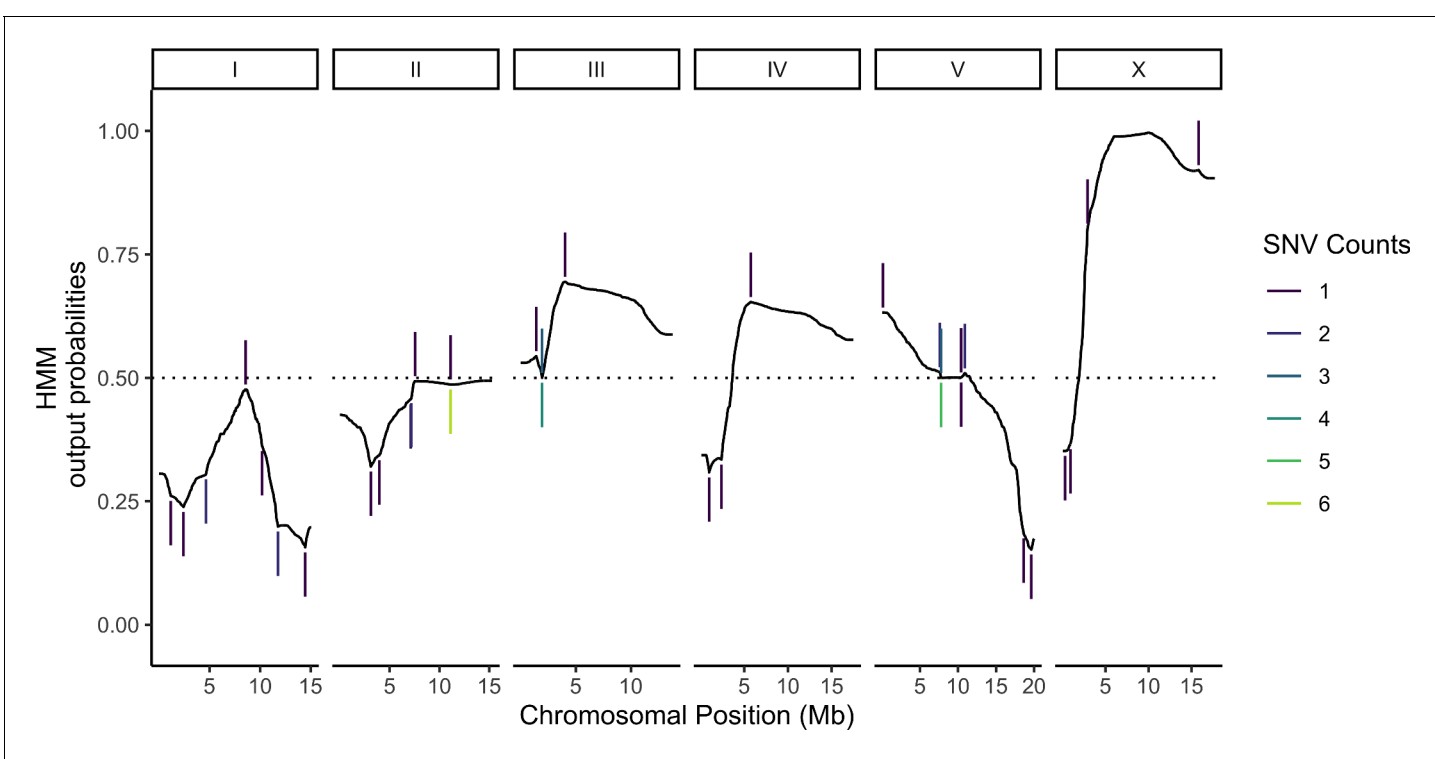

**Figure 2.** Probabilistic genotyping using a hidden Markov model. A cell with the median (69) number of unique genotype-informative single-nucleotide variant (SNV) unique molecular identifier (UMI) counts is shown for illustration. The trace is a summation of the probability of a CB4856 homozygous genotype and half the probability of CB4856 heterozygous genotype at each position. Each vertical line is a count for an SNV, and colors correspond to the count depth. Vertical lines pointing upwards denote counts supporting the CB4856 variant, while lines pointing downwards are counts supporting the N2 variant.

## eQTL mapping in multiple cell types

We mapped 1718 *cis* eQTLs in 1294 genes and 451 *trans* eQTLs in 390 genes at a false discovery rate (FDR) of 10% across the different cell types (*Figure 3A, B*, *Supplementary file 1*—Table S2). The number of eQTLs detected in each cell type was strongly correlated with the number of cells of that type (Spearman's ρ=0.91, p<2.2×10$^{-16}$). In cell types with >1000 cells, we mapped between 52 and 415 eQTLs (*Supplementary file 1*—Table S1). For 1071 of the 1294 genes with a *cis* eQTL (83%), the eQTL was detected in only one cell type. For 208 of the remaining 223 genes (93%), the direction of the eQTL effect was the same in all cell types in which it was detected.

We studied to what degree our *cis* eQTL results were concordant with gene expression differences between the parents. We generated a scRNA-seq dataset from 6721 N2 and 3104 CB4856 cells

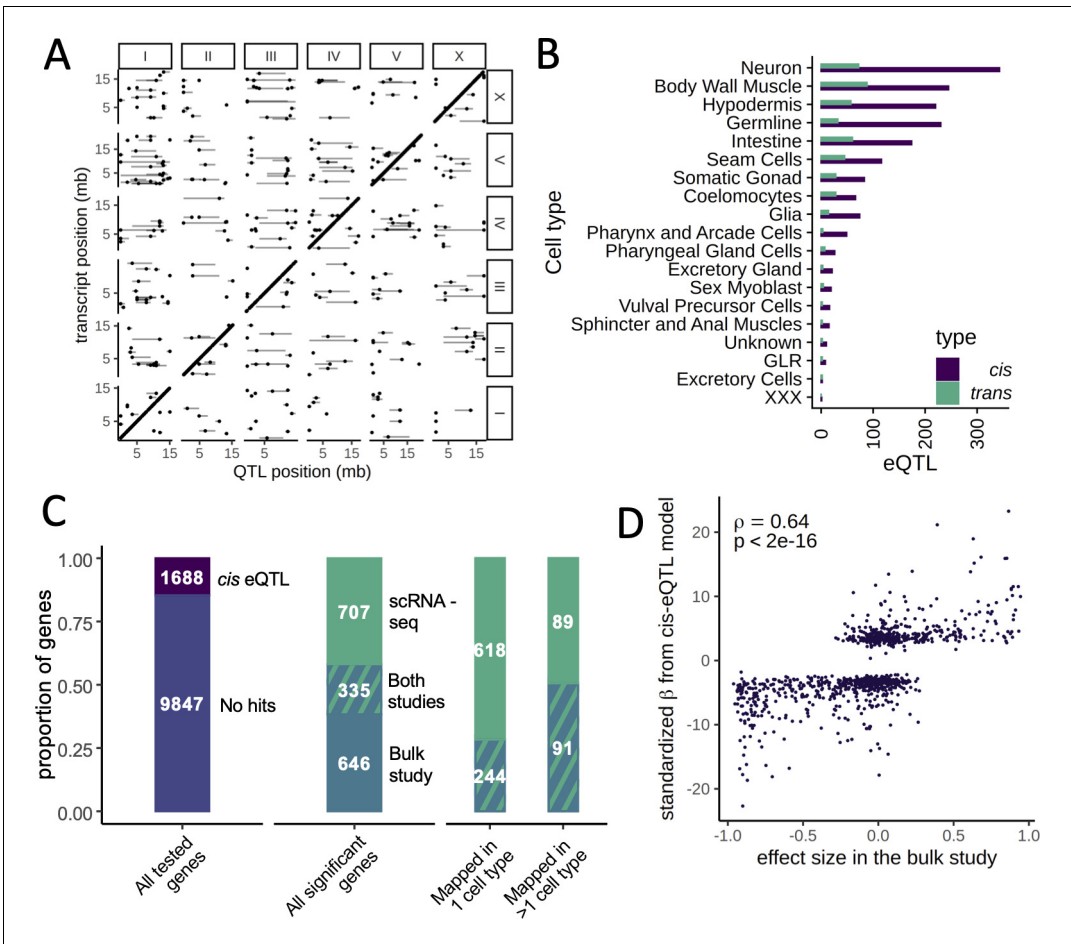

**Figure 3.** Expression quantitative trait loci (eQTL) mapping in cell types. (**A**) A genome-wide map of eQTLs across all cell types is shown. The position of the eQTLs is shown on the x-axis, while the y-axis shows the position of the associated transcripts. Points along the diagonal are *cis* eQTLs (those mapping to nearby genes). (**B**) The number of *cis* and *trans* eQTLs mapped in each cell type. (**C**) The overlap between a previous study that mapped eQTLs in whole worms in a panel of recombinant inbred lines and our dataset. (Left) The proportion of genes with a *cis* eQTL in at least one dataset out of all genes tested. (Middle) Of the 1688 significant *cis* eQTL genes, 355 had a *cis* eQTL in both datasets, representing a highly significant enrichment. (Right) Hits mapped in more than one cell type were more likely to also be found in the whole-worm ('bulk') dataset. (**D**) Quantitative comparison between normalized effect sizes in our dataset and in the whole-worm dataset.

The online version of this article includes the following figure supplement(s) for figure 3:

**Figure supplement 1.** *cis* expression quantitative trait loci (eQTLs) reflect gene expression differences in the parent strains.

**Figure supplement 2.** Correlation between cell-type *cis* expression quantitative trait loci signal and differential gene expression between the parental strains.

and used a classifier trained on the segregant dataset to identify cell types in the parental scRNA-seq dataset. We then carried out a differential expression analysis in each cell type. We found 870 differentially expressed genes (at a greater than twofold change and FDR of 10%), of which 201 (23%) had a *cis* eQTL in the same tissue (odds ratio [OR] = 18.8, p<2.2×10$^{-16}$, Fisher's exact test). In total, 191 of these *cis* eQTL (95%) showed the same direction of effect as the parental difference. Further, the effect sizes of the significant *cis* eQTLs were strongly correlated with the sizes of the parental differences (Spearman's ρ=0.66, p<2.2×10$^{-16}$) (*Figure 3—figure supplements 1* and *2*). These results provide independent support for our *cis* eQTL mapping and show that for a sizable fraction of the genes those *cis* eQTLs are a major cause of differential gene expression between the strains.

## Comparison between bulk and single-cell *cis* eQTL mapping

To investigate the relationship between single-cell and bulk eQTL mapping, we compared our single-cell eQTLs to those previously identified in a panel of 200 recombinant inbred lines (RILs) generated from crossing N2 and CB4856 (*Rockman et al., 2010*). In the bulk study, a large population of whole worms from each RIL was recovered at a late larval stage, L4, and profiled on expression microarrays. We reanalyzed data for 11,535 genes expressed in both datasets and identified 981 *cis* eQTLs in the bulk dataset (at an FDR cutoff of 10%). Despite major differences in experimental design, including the developmental stage of the worms, the overlap with the single-cell *cis* eQTLs was highly significant, with 335 *cis* eQTLs shared between the studies (OR = 7.2, p<2.2×10$^{-16}$, Fisher's exact test) (*Figure 3C*). These shared loci represented 34% of the bulk *cis* eQTLs and 32% of the single-cell *cis* eQTLs. Furthermore, the bulk and single-cell eQTL effect sizes were highly correlated (Spearman's ρ=0.64, p<2.2×10$^{-16}$) (*Figure 3D*). Lastly, single-cell eQTLs detected in multiple cell types were more likely to also be seen in the bulk study: 50% of the genes with *cis* eQTLs detected in multiple cell types were also identified in bulk compared to 28% of the eQTLs detected in only one cell type (OR = 2.58, p=2.1×10$^{-8}$) (*Figure 3C*). This observation suggests that the single-cell eQTL mapping approach improves the power to detect cell-type-specific effects.

## Shared and cell-type-specific *trans* eQTL hotspots

We observed that 90 of the 451 *trans* eQTLs clustered at five hotspots, each containing 12–31 eQTLs (*Figure 4*, *Supplementary file 1*—Table S3). A hotspot on Chr. I was identified independently in both neurons and seam cells; the top associated variant (Chr. I:10890182) was the same for both cell types (*Figure 4A, B*). The other hotspots were identified in the body wall muscle (on Chr. I) (*Figure 4C*), the intestine (on Chr. V) (*Figure 4D*), and neurons (two distinct hotspots on Chr. III) (*Figure 4B*).

To test whether the target genes of these five hotspots are involved in coherent biological processes, we relaxed the FDR threshold to 20%, which increased the number of genes linked to each hotspot to 21–42, and performed Gene Ontology (GO) enrichment analysis (*Supplementary file 1*—Table S4). For three of the hotspots, we found significant enrichments that were consistent with the cell-type specificity of the hotspot. The targets of the hotspot detected in intestinal cells were weakly enriched for genes involved in the innate immune response (FDR-corrected p=0.042), a major role of that tissue (*Pukkila-Worley and Ausubel, 2012*). The targets of the hotspot detected in the body wall muscle were enriched for genes associated with the term *myofilament* (FDR corrected p=6.4×10$^{-8}$), *actin cytoskeleton* (FDR-corrected p=4.2×10$^{-6}$), and related terms. The enrichment was driven by the genes *mup-2, tni-1, tnt-2, mlc-2, mlc-3, lev-11,* and *act-4. mlc-2* and *mlc-3* encode a myosin light chain, and *act-4* encodes an actin protein. *lev-11* encodes a tropomyosin, and *mup-2, tni-1,* and *tnt-2* encode three of the four proteins in *C. elegans* that are expressed in the body-wall muscle and form troponin complexes, highly conserved regulators of muscle contraction (*Ono and Ono, 2004*; *Figure 4—figure supplement 1*).

The targets of the neuronal hotspot on the right arm of Chr. III were enriched for genes involved in *vesicle localization* (FDR-corrected p=7.5×10$^{-3}$), as well as for BMP receptor binding genes (FDR-corrected p=2.8×10$^{-3}$). The latter enrichment was driven by *dbl-1* and *tig-2*, orthologs of human bone morphogenetic protein (BMP) genes BMP5 and BMP8 and ligands of the transforming growth factor beta (TGF-β) pathway (*Gumienny, 2013*). Notably, *dbl-1* was discovered as a gene that regulates body size in *C. elegans* (*Suzuki et al., 1999*), the hotspot peak

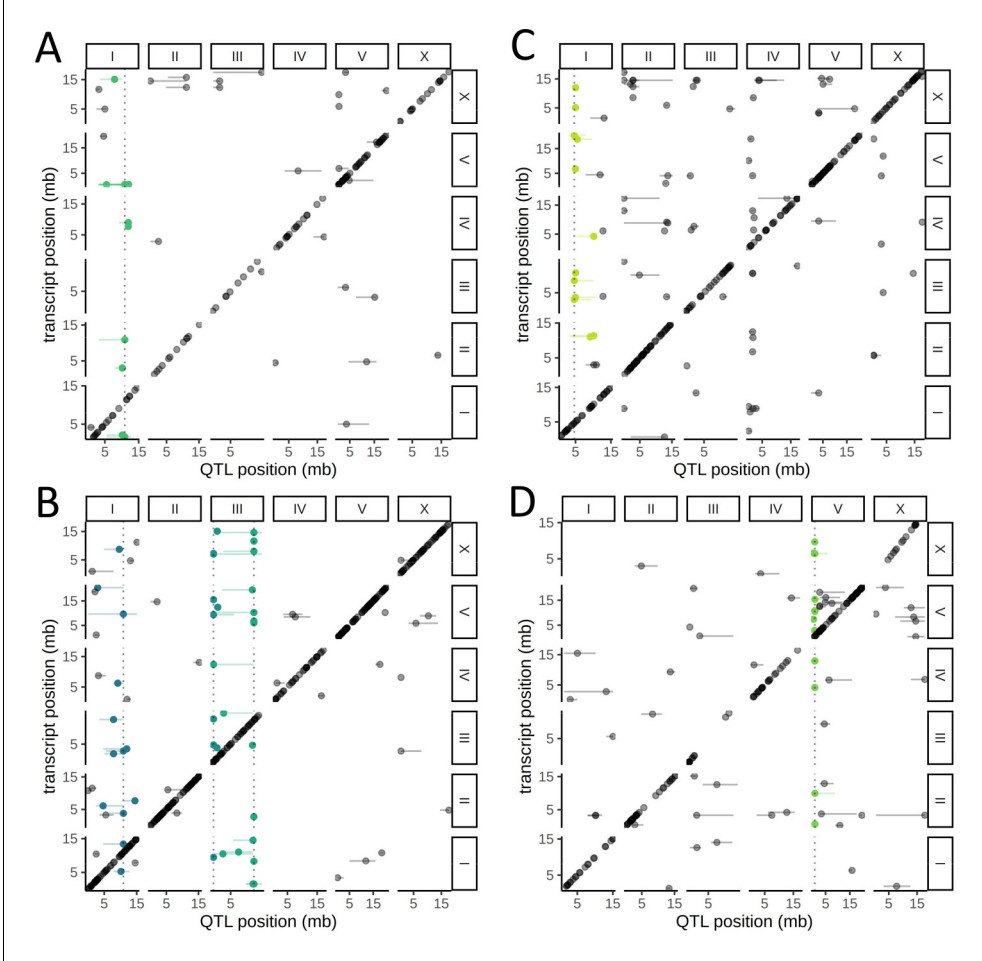

**Figure 4.** Cell-type-specific *trans* expression quantitative trait loci (eQTLs) hotspots. A genome-wide map of eQTLs in seam cells (A), neurons (B), body-wall muscle cells (C), and intestinal cells (D) is shown. The position of the eQTLs is shown on the x-axis, while the y-axis shows the position of the associated transcripts. The dotted line marks the peak position of the hotspot, while targets of each hotspot are colored.

The online version of this article includes the following figure supplement(s) for figure 4:

**Figure supplement 1.** Cell-type expression of genes that form troponin complexes.

marker is located <300 kb from the peak of a QTL we previously identified for body size (*Andersen et al., 2015*), and the corresponding confidence intervals overlap (*Supplementary file 1*—Table S3), suggesting that differential regulation of the TGF-β pathway is involved in variation in body size between N2 and CB4856.

## Cell-specific eQTL effects in the *C. elegans* nervous system

*C. elegans* is a premier model for studying neurobiology at the cellular level, which is aided by its invariant cell lineage and the diverse functions associated with specific individual neurons. Importantly, many of the neurons are highly variable in their gene expression and express specific gene markers (*Hobert et al., 2016*). To identify specific subtypes of neuronal cells, we separately clustered the 12,467 cells identified as neurons and compared the clusters to previous *C. elegans* scRNA-seq datasets, including the recently published *C. elegans* Neuronal Gene Expression Map and Network (CeNGEN) (*Cao et al., 2017*; *Packer et al., 2019*; *Hammarlund et al., 2018*; *Taylor, 2019*; *Supplementary file 1*—Table S5). The neurons fell into 81 distinct clusters, ranging from 17 to 872 cells. We mapped these clusters onto 100 (83%) of the 120 neuronal clusters identified in CeNGEN (*Figure 5—figure supplement 1*). We also identified CEM neurons, which are male

specific and absent from CeNGEN, based on the expression of the marker *cwp-1* (*Portman and Emmons, 2004*).

We mapped *cis* eQTLs in each of the single neuronal subtypes (sn-eQTLs) and identified a total of 163 sn-eQTLs in 132 genes at an FDR of 10% (*Figure 5A*, *Supplementary file 1*—Table S6). Of these, 117 (88%) were identified in only a single neuronal subtype. Functional annotation of sn-eQTLs identified 25 genes involved in signaling (FDR-corrected p=0.047), including 12 genes involved in G-protein coupled receptor signaling (FDR-corrected p=0.047) and eight genes involved in neuropeptide signaling (FDR-corrected p=$9.9\times10^{-3}$).

We compared the sn-eQTLs to those identified when all neurons were analyzed jointly ('pan-neuronal mapping') and found that a sizable fraction of the sn-eQTLs did not have evidence for a pan-neuronal signal: 92 were not identified pan-neuronally at an FDR of 10% and 69 were not identified even at a highly permissive FDR of 50%, suggesting that they exert their effects only in specific

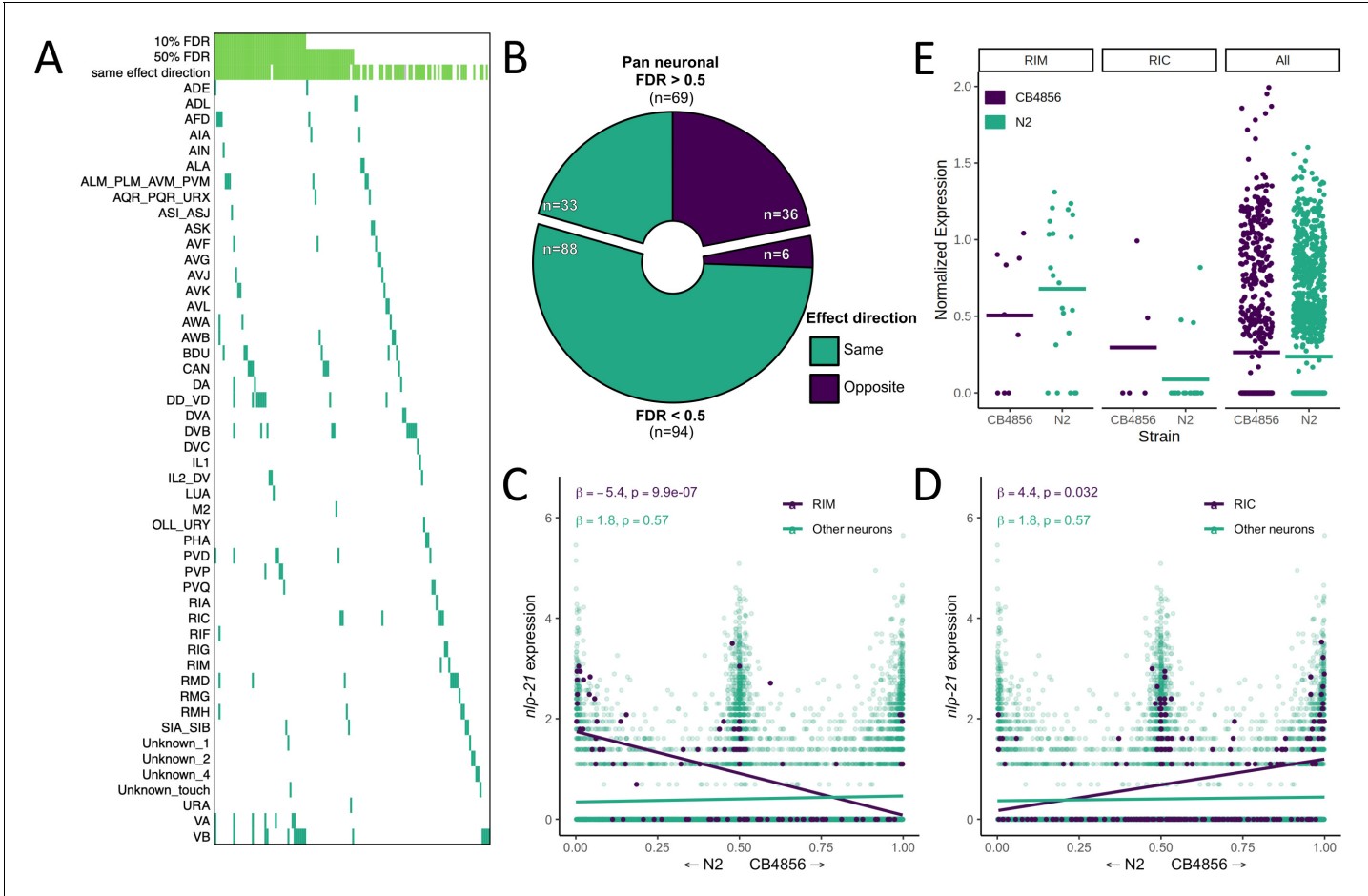

**Figure 5.** Neuron-specific expression quantitative trait loci (eQTL) mapping. (**A**) *cis* eQTLs mapped in single neuronal subtypes (sn-eQTLs) are shown. The top three rows indicate whether the eQTL was mapped pan-neuronally at a 10% false discovery rate (FDR) threshold (row 1), at a 50% FDR threshold (row 2), and whether the sign of the effect estimate ('effect direction') was the same in the pan-neuronal and single-cell mapping (row 3). (**B**) Comparing the effect direction between the sn-eQTL mapping and mapping in a set of neurons excluding the sn-eQTL neuron shows evidence for subtype-specific effects. The number of genes showing the same ('purple') or opposite ('turquoise') effect directions is shown for genes with pan-neuronal FDR > 50% (top) and <50% (bottom). (**C, D**) An eQTL with antagonistic effects in two neurons. Higher expression of the gene *nlp-21* in the RIM neuron is associated with the N2 allele (**C**), while higher expression in the RIC neuron is associated with the CB4856 allele (**D**). In (**C**) and (**D**), a linear fit is shown for illustration. All p-values are FDR-corrected. Read counts were normalized to the number of UMIs in each cell and log-transformed. (**E**) Expression of *nlp-21* in the parental dataset. The direction of effect is concordant between the left panel and (**C**) (RIM neuron) and between the middle panel and (**D**) (RIC neuron). Horizontal lines are averages.

The online version of this article includes the following figure supplement(s) for figure 5:

**Figure supplement 1.** Uniform Manifold Approximation and Projection of 12,468 neurons.

neuronal subtypes (*Figure 5A*). Regardless of statistical significance, pan-neuronal eQTLs should have consistent effect directions across neuronal subtypes, while subtype-specific eQTLs should not. We therefore compared the direction of effect of each sn-eQTL in the subtype in which it was detected with its direction of effect in the set of all neurons excluding that subtype. Among the 69 sn-eQTLs with no signal in the pan-neuronal mapping even at the permissive FDR, the direction of the effect was concordant for 33 and discordant for 36, not significantly different from chance (p>0.5; binomial test), as would be expected if these effects are truly subtype-specific (*Figure 5B*). In contrast, among the 94 that had a pan-neuronal signal at an FDR of 50%, the direction of the effect was concordant for 88 and discordant for only 6 (p<0.000001; binomial test), consistent with differences in detection arising from limited statistical power.

In a striking case, we observed an sn-eQTL in the neuropeptide gene *nlp-21* that showed significant and opposing effects in two neurons (*Figure 5C, D*). In the RIC neuron, higher *nlp-21* expression was associated with the CB4856 allele ($\beta = 4.4$, FDR-corrected p=0.03), while in the RIM neuron, higher *nlp-21* expression was associated with the N2 allele ($\beta = -5.4$, FDR-corrected p=$9.8\times10^{-7}$). In the pan-neuronal mapping, no significant effect is observed for this gene. We identified the RIC and RIM neurons in the parental dataset, and although the small number of cells in each group (35 and 27, respectively, with only 9 and 5 of them from CB4856) was insufficient for statistical testing, the directions of the differences agreed with the eQTL effects (*Figure 5E*). These results provide direct evidence that eQTLs can be specific down to the cellular level.

## Discussion

We used scRNA-seq to map eQTLs in *C. elegans* across cell types in a single one-pot experiment. Earlier scRNA-seq eQTL mapping studies were limited in sample size to at most ~100 individuals, but nevertheless highlighted the potential of this approach to identify cell-type (*van der Wijst et al., 2018*) and developmental (*Cuomo et al., 2020*) eQTLs, as well as loci affecting expression variance (*Sarkar et al., 2019*). Our novel approach allowed us to map eQTLs in tens of thousands of genotypically distinct individuals and enabled detection of both *cis* and *trans* eQTLs, as well as resolution of their effects down to the level of specific cells.

One of the major factors affecting gene expression studies is variation resulting from uncontrolled environmental differences between individuals that are grown or processed separately. By using scRNA-seq, we were able to process all individuals jointly. After the initial parental cross, all subsequent steps carried out over the course of five *C. elegans* life cycles (3 weeks) were performed in bulk, limiting any confounding environmental factors. To minimize the influence of genotype on development, we synchronized the worms at the first larval stage, L1, and collected samples at the L2 stage, limiting the time for differences to accumulate post synchronization. Even careful synchronization is not expected to completely remove the effects of genetic variation on developmental timing, and such variation can be combined with gene expression time-course data collected during development to increase the power of eQTL mapping and study the developmental dynamics of eQTLs (*Francesconi and Lehner, 2014*). This raises the possibility that future scRNA-seq studies of *C. elegans* across developmental stages would open the door to a similar analysis of our single-cell eQTL dataset.

Previous work suggested the existence of cell-type-specific eQTL hotspots in *C. elegans* based on the expression patterns of hotspot targets (*Francesconi and Lehner, 2014*). We discovered three hotspots that are cell-type specific, with targets that are involved in core functions performed by these cell types. Recently, eQTL hotspots have been identified in human blood cells (*Yao et al., 2017*; *Kolberg et al., 2020*), as well as in cell lines (*Brynedal et al., 2017*). These results suggest that hotspot and *trans* eQTL discovery is facilitated by expression studies that can distinguish cell types and point to a larger role of hotspots in the genetics of gene expression in animals.

The extent to which eQTL hotspots play a role in determining biological phenotypes is unknown, although several examples have been described in animals and plants (*Albert and Kruglyak, 2015*; *Brynedal et al., 2017*; *Pang et al., 2019*; *Andersen et al., 2014*; *Orozco et al., 2012*). One of the neuronal hotspots we identified in this study affects the expression of members of the TGF-β pathway, which has been linked to the control of body

size in *C. elegans* (*Gumienny, 2013*). Remarkably, our previous study mapped a QTL for body size in a cross between N2 and CB4856 to the same region (*Andersen et al., 2015*). Ultimately, resolving the physiological impact of eQTL hotspots, including the ones in this study, will require fine-mapping to identify the underlying causal variants. A systematic approach for fine-mapping eQTL hotspots has recently been developed in yeast (*Lutz et al., 2019*); future advances may enable this to be carried out in animals as well.

A comparison of the single-cell *cis* eQTLs to those mapped in a previous whole-worm eQTL study from our laboratory showed a highly significant overlap despite major differences in experimental design. These results join accumulating evidence that a sizable fraction of *cis* eQTLs have robust, consistent effects (*GTEx Consortium, 2020*; *Smith and Kruglyak, 2008*) and show that many of the effects are conserved across worm development. The strong overlap of *cis* eQTLs mapped by scRNA-seq and whole-worm analysis also suggests that the effect of many *cis* eQTLs is conserved across cell types. This observation suggests that modeling strategies that borrow power among cell types could increase the power to detect eQTLs. In addition, our approach for *cis* eQTL discovery could be supplemented with tests for allele-specific expression (ASE). Looking forward, our method enables scaling up the number of studied individuals, and hence increasing statistical power, simply by sequencing a larger number of cells. Thus, the increasing throughput of single-cell technologies and sequencing platforms will enable future work to study cell-type specificity of *cis* and *trans* eQTLs in greater detail.

Lastly, we discovered *cis* eQTLs that act in single subtypes of *C. elegans* neurons, including many that were not found when all neurons were analyzed jointly. Importantly, we also discovered an eQTL that influences expression of the gene *nlp-21* in opposing directions in two different neurons. Such antagonism could result from context-dependent effects of the same variant or from different variants acting in each cell. Our study joins a growing body of work that identified eQTLs that are specific to cell types (*Fairfax et al., 2012*; *Ishigaki et al., 2017*; *Donovan et al., 2020*; *Kim-Hellmuth et al., 2020*; *van der Wijst et al., 2018*; *Zhang et al., 2018*; *Westra et al., 2015*), as well as developmental (*Cuomo et al., 2020*; *Francesconi and Lehner, 2014*) and environmental (*Orozco et al., 2012*; *Fairfax and Knight, 2014*; *Kim-Hellmuth et al., 2017*) contexts. Distinct genetic effects are found across cell types and conditions, and large-scale study of such effects is therefore crucial for gaining a comprehensive understanding of regulatory variation.

# Materials and methods

**Key resources table**

| Reagent type (species) or resource | Designation | Source or reference | Identifiers | Additional information |
|---|---|---|---|---|
| Strain, strain background (*Caenorhabditis elegans*) | N2 fog-2(q71) V; hsp-90p::GFP II | *Burga et al., 2019* | QX2314 | A *C. elegans* strain carrying a *fog-2* and a daf-21p::GFP reporter in the N2 genetic background |
| Strain, strain background (*Caenorhabditis elegans*) | CB4856 fog-2(kah89) | *Burga et al., 2019* | PTM299 | A *C. elegans* strain carrying a *fog-2* mutation in the PTM299 genetic background |
| Commercial assay or kit | Chromium Single Cell 3' v3 | 10x Genomics | CG000201 | |
| Software, algorithm | Monocle3 | *Cao et al., 2019* | | Available at: https://cole-trapnell-lab.github.io/monocle3/ |
| Software, algorithm | eQTL mapping code | This paper | | Available at: https://github.com/joshsbloom/single_cell_eQTL; copy archived at swh:1:rev:321e29c20cecab726426053bc5a6160b66284691; *Ben-David, 2021a* |

*Continued on next page*

*Continued*

| Reagent type (species) or resource | Designation | Source or reference | Identifiers | Additional information |
|---|---|---|---|---|
| Software, algorithm | Code to reconstruct figures and calculate all numbers in the manuscript | This paper | | Available at: https://github.com/eyalbenda/worm_sceQTL; copy archived at; swh:1:rev:e30eb7d2a393459b2367dfe22c07966e44364f20; *Ben-David, 2021b* |

## *C. elegans* culturing

*C. elegans* strains were cultured at 20°C using standard conditions with the exception that the agar in the nematode growth media (NGM) was replaced with a 4:6 mixture of agarose and agar (NGM +agarose) to prevent burrowing of the CB4856 strain. Parental strains used were QX2314 (N2 *fog-2* (q71) V; *hsp-90p*::GFP II) and PTM299 (CB4856 *fog-2*(kah89)). Large segregant panels were generated as before (*Burga et al., 2019*). Briefly, 500 L4 males from PTM299 and 500 L4 hermaphrodites from QX2314 were seeded on a plate for 30 hr, and gravid worms and eggs were collected and bleached. Eggs were synchronized to L1 larvae for 24 hr and seeded on 10 cm NGM+agarose plates. In each generation, gravid worms were bleached, their progeny synchronized for 24 hr, and seeded. The entire process was repeated up to F4, with 3–4 days per generation.

## Cell extraction and sequencing

In total, 192,000 F4 were seeded on four 10 cm NGM+agarose plates seeded with OP50. L2 were recovered after 24 hr, and staging was validated under a stereomicroscope. L2 cell dissociation was carried out as previously described (*Zhang et al., 2011*), implementing modifications from a later study (*Kaletsky et al., 2016*), as well as our own. Worms were recovered off the plates and washed three times in M9. Lysis was then done with an SDS-DTT solution (200 mM DTT, 0.25% SDS, 20 mM HEPES, pH 8.0, 3% sucrose) in a hula mixer set on low speed to prevent worms from settling. The lysate was observed under the stereoscope every 2 min, and lysis was stopped when a blunted head shape appeared in the majority of worms (*Kaletsky et al., 2016*), after ~4 min. Worms were then washed quickly three times in 1 ml of M9, and two additional times in 1 ml of egg buffer (118 mM NaCl, 48 mM KCl, 2 mM CaCl$_2$, 2 mM MgCl$_2$, 25 mM HEPES, pH 7.3, osmolarity adjusted to 340 mOsm with sucrose). Worms were then resuspended in 0.5 ml of 20 mg/ml Pronase E that was freshly prepared in L15 media supplanted with 2% fetal bovine serum (L15-FBS) and adjusted to 340 mOsm with sucrose. Worm dissociation was done by continuous pipetting on the side of the tube and monitored every 2–3 min on a microscope equipped with a ×40 phase contrast objective lens. Dissociation was stopped when few intact worms remained and a high density of cells was visible. Then, 0.5 ml of L15-FBS was added to stop the reaction, and the lysate was spun for 6 min at 500 g at 4°C. The cell pellet was resuspended in PBS (PBS was adjusted to 340 mOsm with sucrose). Cell suspension was spun for 1 min in 100 g at 4°C to remove remaining undigested worms, counted, and diluted to 1 M cells/ml in osmolarity-adjusted PBS, and loaded directly onto five lanes of 3′ Chromium single-cell RNA-sequencing flow cells (10x Genomics), targeting 10,000 cells on each lane. Library prep was carried out according to manufacturer's protocol. Prepared libraries were sequenced together on an S4 lane of Novaseq 6000. A paired-end 2 × 150 run was done to maximize the recovery of single-nucleotide variants (SNVs). In all downstream processing, each of the five 3′ Chromium lanes processed concurrently was treated as a separate 'batch', and lane identity corresponds to the 'batch' identity for the rest of the methods.

## Single-cell RNA-sequencing data processing

Raw sequencing reads were analyzed using *CellRanger* (version 3.0.2). We used a gene transfer format file that was corrected for misannotation of 3′ untranslated regions that was generated in a previous study (*Packer et al., 2019*). *C. elegans* cell types differ widely in the number of UMIs that are recovered using scRNA-seq. Therefore, a simple UMI cutoff, as is commonly used, may be biased for cell types with more UMIs. We therefore implemented an iterative pipeline to recover clusters of *bona fide* cells and remove cell doublets as well as degraded cells. We took 20,000 cells with the

most UMIs in each cluster (twice the targeted number of cells, 100,000 overall), and processed them in *Monocle* (version 3) (*Qiu et al., 2017*). Default parameters were used, with the exception that 100 dimensions were used for reduction, and batch was added as a covariate. Leiden clustering identified a total of 154 clusters, and we used the *top_markers* function in *Monocle* to identify the genes upregulated in each. We then removed clusters whose top genes included any ribosomal genes or the mitochondrial genes *ndfl-4*, *nduo-6*, *atp-6*, *ctc-2*, *ctc-3*, *ctc-1*, which we noticed were usually found together as the most upregulated genes in clusters that did not specifically express any known markers for *C. elegans* cell types. This removed a total of 30,980 cells (31%). For the remaining 69,020 cells, raw counts were processed using the R package *SoupX* to reduce ambient RNA contamination (*Young and Behjati, 2020*). We then normalized, reduced dimensions, and clustered the background-corrected cell profiles in *Monocle* using the same parameters as above.

To annotate cell types, we used the markers described in a previous study (Table S12 in *Packer et al., 2019*) that reanalyzed a previous L2 single-cell dataset (*Cao et al., 2017*). Our cell-type annotation corresponds to the 'UMAP' column in that table, with the following exceptions: (1) we separated hypodermis from seam cells, somatic gonad from sex muscle cells, and glia from excretory cells since those groups were not clustering together in our data. (2) Cells identified as 'Miscellaneous' in that table were annotated as individual cell-type identifications in our data, with the exception of the sphincter and anal muscles, which were not differentiated from each other in our data. Finally, we re-evaluated our cell-type identifications and filtered cell doublets as well as dead cell or debris that may still contaminate *bona fide* cell-type clusters. We trained a classifier using our manually curated cell-type classifications with a L2-penalized multinomial logistic regression framework, as implemented in the *Scikit-learn* Python package (v0.22) (*Pedregosa, 2011*). We read the raw gene expression matrices into Python using scanpy (v1.4.2). We removed 2582 genes that were expressed in less than 10 cells. The gene expression levels of each cell were corrected so that the total gene expression counts added up to 10,000. Per gene, these corrected counts were normalized using a log(1+x) transformation. To speed up the computation of the multinomial logistic regression, we only used the 2037 genes with a mean expression between 0.0125 and 3, and a minimum dispersion of 0.5. We scaled the gene expression matrix so that the expression level of each gene across cells had a mean of 0 and a variance of 1, after scaling expression values over 20 were set to 20. We fit a multinomial logistic regression model using the scaled gene expression values for the 2037 highly variable genes from the complete set of 69,020 cells to obtain an estimate for the inverse regularization strength (C). Using the estimated C of $7.74 \times 10^{-04}$, we performed fivefold cross-validation to estimate the probability that each cell belongs to one of the manually curated cell-type classifications. Any cell with a probability higher than 0.2 of belonging to two or more cell types (9198) was classified as a doublet. Any cell that did not belong to a cell type with probability $\geq$0.4 (5547) was classified as low quality. In total, we removed 11,398 cells that were classified as a doublet or low quality. We removed an additional 2114 cells classified as Neurons as described in the section 'Neuronal cell-type classification'. For the remaining final list of 55,508 cells, we used the output of the classifier as the final cell-type classification. The final classification is shown in *Figure 1*. For display purposes, the plot in *Figure 1* was generated by rerunning umap on the finalized dataset with *euclidean* distance metric and *umap.min_dist* = 0.5, resulting in a more compressed visualization of the dataset.

## Estimating the number of unique genotypes

We estimated the number of unique genotypes in two approaches. First, we noted that calculating the number of expected unique genotypes is akin to the well-known 'Birthday problem' in statistics. Given C cells sampled from I individuals, the expected number of cells with a unique genotype is $C(1-1/I)^{C-1}$. Assuming 50–90% of worms were successfully dissociated (a conservative range), we expect 31,134–40,257 unique genotypes.

We also calculated an empirical measurement for collisions. We calculated the genotype correlation matrix of the cells and counted the cells with a maximal correlation with another cell is higher than 0.9. This is presented for each cell type in *Supplementary file 1—Table S1*.

## Single-nucleotide variant counting

We used a list of SNVs we previously curated for CB4856 compared to the N2 reference (*Burga et al., 2019*). We derived genotype informative UMI counts for N2 and CB4856 variants using *Vartrix version 1.0* (https://github.com/10XGenomics/vartrix) directly on the output of *Cell-Ranger*. To reduce SNV counts that result from SNVs in the ambient RNA background, we only kept SNVs that resided in genes with positive counts in the background-corrected matrix.

## Genotype inference using a HMM

We set up a HMM to infer the genotypes of the recombinant progeny (*Broman, 2005*; *Arends et al., 2010*). The HMM is used to calculate the probability of underlying genotypes for each individual and requires three components: (1) prior probabilities for each of the possible genotypes, (2) emission probabilities for observing variant-informative reads given each of the possible genotypes, (3) and transition probabilities – the probabilities of recombination occurring between adjacent genotype-informative sites.

For the autosomal chromosomes, we defined prior genotype probabilities as 0.25 for homozygote N2, 0.5 for heterozygote CB4856/N2, and 0.25 for homozygote CB4856. For the sex chromosome, we defined prior genotype probabilities as 0.44 for homozygotes N2, 0.44 for the heterozygote CB4856/N2, and 0.11 for homozygote CB4856. These values were chosen because to generate the segregant population N2 hermaphrodites were crossed to CB4856 males and thus contributed twice as many X chromosomes to the progeny as CB4856.

Emission probabilities were calculated as previously described for low-coverage sequencing data (*Dodds et al., 2015*; *Bilton et al., 2018*) under the assumption that the observed counts of reads for both possible variants (Y) at a genotype-informative site (g) arise from a random binomial sampling of the alleles present at that site and that sequencing errors (e) occur independently between reads at a rate of 0.002:

$$p(Y|g = NN) = \binom{D}{r}(1-e)^r(1-(1-e))^{D-r}$$

$$p(Y|g = NC) = \binom{D}{r}\frac{1}{2}^D$$

$$p(Y|g = CC) = \binom{D}{r}(e)^r(1-e)^{D-r}$$

where (D) is the total read depth at a genotype-informative site for a given individual, (r) is the total read depth for the N2 variant at that site, and N represents the N2 variant and C represents the CB4856 variant.

Transition probabilities were derived from an existing N2 × CB4856 genetic map (*Rockman and Kruglyak, 2009*). We linearly interpolated genetic map distances from the existing map to all genotype-informative sites in our cross progeny. We scaled these genetic map distances, multiplying them by a factor of 0.4, to account for the fact that the previous genetic map was built using 10 generations of intercrossing, whereas progeny from our cross are derived from four generations of intercrossing (*Rockman and Kruglyak, 2008*).

For QTL mapping, we used an additive coding, summing the probability that the genotype was homozygote CB4856 with one half the probability that the genotype was heterozygote N2/CB4856.

## eQTL mapping

Genotype probabilities were standardized, and markers in very high LD (r > 0.9999) were pruned. This LD pruning is approximately equivalent to using markers spaced 5 centimorgans (cm) apart. For each transcript, we counted the number of cells for which at least one UMI count was detected in each cell type. Transcripts with non-zero counts in at least 20 cells in a cell type were considered expressed in that cell type and used for downstream analyses.

As has been previously described for droplet scRNA-seq, counts of UMIs can be adequately parameterized by a gamma-Poisson distribution, which is also known as the negative binomial

distribution (*Svensson, 2020*). Thus we used a negative binomial regression framework for eQTL mapping here. We also note that simpler approaches using log(counts+1) with ordinary least squares behave pathologically, especially in regard to behavior with multiple partially correlated covariates, and simulations (not shown) showed such models lead to inflated false-positive rates.

For each expressed transcript in each cell type, we first fit the negative binomial generalized linear model:

$$E[Y] = \mu \tag{1}$$

$$Var(Y) = \mu + \frac{1}{\theta}\mu^2 \tag{2}$$

$$\mu = exp(\beta_i + X_t\beta_t + X_b\beta_b + X_c\beta_c) \tag{3}$$

which has the following log-likelihood:

$$l(\beta,\theta) = -\sum_{n=1}^{N}[(y_n+\theta)log(\mu_n+\theta) - y_nlog(\mu_n) + log(|\Gamma(y_n+1)|) - \\ \theta log(\theta) + log(|\Gamma(\theta)|) - log(|\Gamma(\theta+y_n)|)] \tag{4}$$

where Y is a vector of UMI counts per cell, $X_t$ is a vector of the log(total UMIs per cell) and controls for compositional effects, $X_b$ is an indicator matrix assigning cells to batches, and $X_c$ is the vector of standardized genotype probabilities across cells for the closest genotypic marker to each transcript from the pruned marker set. In addition, β is a vector of estimated coefficients from the model, $\mu_n$ is the expected value of Y for a given cell n, N is the total number of cells in the given cell type, and θ is a negative binomial overdispersion parameter. Model parameters were estimated using iteratively reweighted least squares as implemented in the *negbin.reg* function in the *Rfast2* R package. If the model did not converge, model parameters were estimated with the *gam* function in the *mgcv* package (*Wood, 2017*), which opts for certainty of convergence over speed. We note that due to the computational burden of fitting so many generalized linear models (GLMs) in the context of sc-eQTL mapping, we chose to estimate θ once for each transcript in each cell type for this model and use that estimate of θ in the additional models for that transcript within the cell type, as described below. This approach is conservative as the effects of unmodeled factors (e.g., *trans* eQTLs) will be absorbed into the estimate of overdispersion, resulting in larger estimated overdispersion ($\frac{1}{\theta}$) and lower model likelihoods. Computational approaches that re-estimate θ for each model, jointly model all additive genetic effects, or regularize θ across models and transcripts (*McCarthy et al., 2012*) may further increase statistical power to identify linkages.

To evaluate the statistical significance of *cis* eQTLs, a likelihood ratio statistic, -2($l_{nc}$-$l_{fc}$), was calculated comparing the log-likelihood of this model described above (lfc) to the log-likelihood of the model, where β is re-estimated while leaving out the covariate $X_c$ for the *cis* eQTL marker (lnc). A p-value was derived under the assumption that this statistic is $X^2$ distributed with one degree of freedom. This p-value was used for the evaluation of significance of *cis* eQTLs for the neuronal subtypes. Within each neuronal subtype, FDR-adjusted p-values were calculated using the method of *Benjamini and Hochberg, 1995*. For the other cell types (with typically much larger cell numbers) and the genome-wide scans for eQTLs, a permutation procedure was used to calculate FDR-adjusted p-values and is described further below.

For each expressed transcript in each cell type, we also scanned the entire genome for eQTLs, enabling detection of *trans* eQTLs. A similar procedure was used as for *cis* eQTL except that *Equation (3)* was replaced with

$$\mu = exp(\beta_i + X_t\beta_t + X_b\beta_b + X_g\beta_g) \tag{5}$$

where $X_g$ is a vector of the scaled genotype probabilities at the gth genotypic marker, and the model is fit separately, one at a time, for each marker across the genome for each transcript. A likelihood ratio statistic for each transcript, within each cell type, for each genotypic marker is calculated by comparing this model to the model where β is re-estimated while leaving out the covariate $X_g$. The likelihood ratio statistic was transformed into a logarithm of the odds (LOD) score by dividing it

by $2\log_e(10)$. We also used functions in the *fastglm* R package for this scan, again re-using estimates of θ obtained as described above for each transcript for each cell type. For each transcript and each chromosome, QTL peak markers were identified as the marker with the highest LOD score. The 1.5 LOD-drop procedure was used to define approximate 95% confidence intervals for QTL peaks (*Dupuis and Siegmund, 1999*).

FDR-adjusted p-values were calculated for QTL peaks. They were calculated as the ratio of the number of transcripts expected by chance to show a maximum LOD score greater than a particular LOD threshold vs. the number of transcripts observed in the real data with a maximum LOD score greater than that threshold, for a series of LOD thresholds ranging from 0.1 to 0.1+the maximum observed LOD for all transcripts within a cell type, with equal-sized steps of 0.01. Per chromosome, the number of transcripts expected by chance at a given threshold was calculated by permuting the assignments of segregant identity within each batch relative to segregant genotypes, calculating LOD scores for all transcripts across the chromosome as described above, and recording the maximum LOD score for each transcript. In each permutation instance, the permutation ordering was the same across all transcripts. We repeated this permutation procedure 10 times. Then, for each of the LOD thresholds, we calculated the average number of transcripts with maximum LOD greater than the given threshold across the 10 permutations. We used the *approxfun* function in R to interpolate the mapping between LOD thresholds and FDR and estimate an FDR-adjusted p-value for each QTL peak (*Albert et al., 2018*).

The same procedure was performed for *cis* eQTL analysis, with the difference being that the expected and observed number of transcripts at a given LOD threshold were calculated only at the marker closest to the transcript. We note that, as expected, Benjamini and Hochberg-adjusted p-values and FDR-adjusted p-values from this permutation procedure for *cis* eQTLs were nearly identical.

## scRNA-seq in the parental strains

The parental QX2314 and PTM299 strains were grown separately for four generations on 10 cm plates, with recurrent cycles of bleaching and synchronization as was done for the segregant population. For single-cell preparation, synchronized L1 from both strains were seeded together in equal numbers on 10 cm plates, and they were processed together from that point onwards, to limit any environmental effects. We believe that differences in efficiency of the cell preparation procedure between N2 and CB4856 could explain the imbalanced representation in the final dataset (6721 N2 and 3104 CB4856 cells). We took advantage of the different parental genotypes when processing the cells and called cells as those with at least 50 SNV counts supporting one genotype and less than 50 supporting the other. Cell-type identification was automated by using the logistic regression model trained on the segregants that is discussed above. Differential expression analysis was carried out using the *DEsingle* R package in each cell type, as well as globally in all cells combined (*Miao et al., 2018*).

To compare differential expression results with our *cis* eQTL results, we first normalized the effect size of each *cis* eQTL by its standard error. Those were used directly in comparisons done within each cell type. To compare with global differential expression, those standardized effects were combined across all cell types in which an eQTL was identified using Stouffer's weighted-Z method (*Whitlock, 2005*).

## Processing whole-worm eQTL data

Microarray genotype and gene expression data for our published expression QTL data were acquired from the gene expression omnibus (*Rockman et al., 2010*). Probe sequences were realigned to the WBcel235 transcriptome using BWA, and uniquely mapping probes were used. Expression probes that were present in less than 2/3 of the sample were removed. The genotype and expression matrices were standardized. To map eQTLs, we calculated the Pearson correlation between each probe and every genotype. Correlation coefficients were transformed to LOD scores using $-n \cdot \frac{\ln(1-R^2)}{2\ln(10)}$. To assess significance and account for multiple testing, we permuted the sample identities 100 times and calculated the average number of transcripts with an identified eQTL at different LOD scores. We compared these results to the unpermuted LOD scores to estimate the FDR and selected a cutoff corresponding to a rate of 10% (LOD = 4.2), equivalent to the single-cell mapping. *cis* eQTLs were derived by calculating the Pearson correlation between transcript expression

and the normalized genotypes in the variant nearest to a given transcript, transforming to LOD score and comparing against the global threshold.

## Hotspot analysis

To discover hotspots, we split the genome into 130 bins of 5 cm each. We then counted the number of eQTLs in each bin identified in each cell type (applying a 10% FDR significance threshold), after removing all *cis* linkages. *Cis* linkages were defined here as those where the transcribed gene falls within the 95% eQTL confidence interval range extended by 1 MB on both sides. A bin was considered to have an excess of linkages if the number of linkages exceeded the number expected by chance from a Poisson distribution, given the average number of linkages per bin for that cell type and a Bonferroni correction for the total number of bins (p<3.8e-4) (*Smith and Kruglyak, 2008*). The *findpeaks* function in the *pracma* R package was used to identify peak hotspot bins and prevent identifying sets of adjacent bins as hotspots.

For GO analysis, we identified hotspot targets using the same procedure above, but relaxed the significance threshold to 20% FDR. We then used the R package *topGO* to identify enriched terms, with the genes expressed in the cell type used as background.

## Neuronal cell-type classification

Neuronal classification was carried out using a combination of available *C. elegans* scRNA-seq datasets, including a published L2 dataset (*Cao et al., 2017*; *Packer et al., 2019*), and the CeNGEN project (*Taylor, 2019*). Neuronal cells were processed separately using *monocle3* with default parameters, with the exception that 100 dimensions were specified for the *preprocess_cds* step. The analysis was carried out in two passes. In the first pass, we processed all cells identified by our classifier as neurons. Following Leiden clustering, we removed 2114 cells that were in clusters whose top genes were mostly mitochondrial and ribosomal genes, similar to the analysis described above for the global dataset. We then processed the pruned dataset in *monocle3* as described above. To annotate the final neuronal clusters, we first used the list of marker genes from two previous publications (*Packer et al., 2019*; *Hammarlund et al., 2018*) to derive candidate clusters that uniquely express marker genes. We next used the *top_markers* function in *monocle3* to identify upregulated genes in each cluster compared to the rest. These were compared with the data available in the online SCeNGEN *Shiny* application (https://cengen.shinyapps.io/SCeNGEA) for the candidate cluster. The full list of genes used for classification is found in *Supplementary file 1*—Table S5. In the final dataset, clusters Unknown_1 - Unknown_4 are of unknown identity and do not correspond to the clusters of the same name in CeNGEN, while the clusters Unknown_touch and Unknown_glut_2 do correspond to cell clusters of the same names in CeNGEN.

sn-eQTL analysis sn-eQTL mapping is described above (section 'eQTL mapping'). GO annotation of genes with sn-eQTL was done in *topGO*, with the genes expressed in neurons (determined using the criteria for inclusion in eQTL mapping) used as background. A heatmap was plotted using the *ComplexHeatmap* package (*Gu et al., 2016*). To determine the consistency in effect direction between the sn-eQTL neuron and the rest of the neurons, we repeated the eQTL mapping, aggregating cells from all neuronal cell types but omitting the neuron with the sn-eQTL. The RIC and RIM neurons in the parental datasets were identified using the same gene markers used in the segregant eQTL dataset, as described in *Supplementary file 1*—Table S5.

## Acknowledgements

The work was supported by the Howard Hughes Medical Institute and NIH grants R01 HG004321 (LK) and K99 HG010369 (EB-D). *Figure 1A* was created with Biorender.com. We thank Michael Mashock, Marco De Simone, and other members of the Technology Center for Genomics and Bioinformatics (TCGB) at UCLA for preparing and sequencing 10x 3′ chromium single-cell libraries.

## Additional information

### Funding

| Funder | Grant reference number | Author |
|--------|------------------------|--------|
| National Human Genome Research Institute | K99-HG010369 | Eyal Ben-David |
| National Human Genome Research Institute | R01-HG004321 | Leonid Kruglyak |
| Howard Hughes Medical Institute | Investigator award | Leonid Kruglyak |

The funders had no role in study design, data collection and interpretation, or the decision to submit the work for publication.

### Author contributions

Eyal Ben-David, Conceptualization, Formal analysis, Funding acquisition, Investigation, Methodology, Writing - original draft, Writing - review and editing; James Boocock, Conceptualization, Formal analysis, Investigation, Methodology, Writing - review and editing; Longhua Guo, Stefan Zdraljevic, Investigation; Joshua S Bloom, Conceptualization, Formal analysis, Supervision, Investigation, Methodology, Writing - review and editing; Leonid Kruglyak, Conceptualization, Supervision, Funding acquisition, Writing - original draft, Writing - review and editing

### Author ORCIDs

Eyal Ben-David (iD) https://orcid.org/0000-0003-0514-0400
Joshua S Bloom (iD) http://orcid.org/0000-0002-7241-1648
Leonid Kruglyak (iD) https://orcid.org/0000-0002-8065-3057

### Decision letter and Author response

Decision letter https://doi.org/10.7554/eLife.65857.sa1
Author response https://doi.org/10.7554/eLife.65857.sa2

## Additional files

### Supplementary files

• Supplementary file 1. Tables S1–S6.
• Transparent reporting form

### Data availability

Raw sequencing data are available under NCBI BioProject PRJNA658829.

The following datasets were generated:

| Author(s) | Year | Dataset title | Dataset URL | Database and Identifier |
|-----------|------|---------------|-------------|-------------------------|
| Ben-David E, Boocock J, Guo L, Zdraljevic S, Bloom JS, Kruglyak L | 2020 | Whole-organism mapping of the genetics of gene expression at cellular resolution | https://www.ncbi.nlm.nih.gov//bioproject/PRJNA658829 | NCBI BioProject, PRJNA658829 |
| Ben-David E, Boocock J, Guo L, Zdraljevic S, Bloom JS, Kruglyak L | 2020 | Whole-organism mapping of the genetics of gene expression at cellular resolution | https://trace.ncbi.nlm.nih.gov/Traces/sra/?study=SRP279842 | NCBI Sequence Read Archive, SRP279842 |

The following previously published dataset was used:

| Author(s) | Year | Dataset title | Dataset URL | Database and Identifier |
|---|---|---|---|---|
| Rockman M, Skrovanek S, Kruglyak L | 2012 | QX recombinant inbred advanced intercross lines of *C. elegans* | https://www.ncbi.nlm.nih.gov/geo/query/acc.cgi?acc=GSE23857 | NCBI Gene Expression Omnibus, GSE23857 |

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
