## [Decision Letter]

**Acceptance summary:**

Genetic variation and cell type specific regulation can intersect to influence quantitative traits. However, finding these cell type specific eQTLs is complicated by the factorial nature of these experiments. In this work, the authors use a pooled transcriptome/genotyping approach with single cell sequencing to begin obtaining a broad overview of the genetic architecture of genotype x cell type interactions.

**Decision letter after peer review:**

Thank you for submitting your article "Whole-organism eQTL mapping at cellular resolution with single-cell sequencing" for consideration by *eLife*. Your article has been reviewed by three peer reviewers, one of whom is a member of our Board of Reviewing Editors, and the evaluation has been overseen by Patricia Wittkopp as the Senior Editor. The following individual involved in review of your submission has agreed to reveal their identity: Ewan Birney (Reviewer #3).

Essential Revisions:

1) Clarify and explain the genotyping information/approach that is largely in the supplement. Each reviewer has a comment along these lines.

2) Temper some claims in the Introduction and Discussion per the reviewers suggestions.

Reviewer #1 (Recommendations for the authors):

The concept of using transcriptomes to simultaneously genotype and phenotype samples is actually a somewhat old concept first established in plants in the era of microarrays. I'm not sure if it is appropriate to not cite the original literature establishing this concept. Some acknowledgment of past developments should be provided.

Introduction – Is it really valid to say that eQTLs have been proposed to underly genetic associations? Seems like there is a ton of validation studies in Yeast, *C. elegans*, *Drosophila*, Maize and Arabidopsis that move this beyond the "proposed" stage.

Results – the authors report that the # of eQTLs is correlated with the number of cells of that type in the sequencing data. This suggests that what is actually happening is the # of cells is correlated with the number of independent individuals represented. It would help if the authors could use the genotyping information to estimate the number of individuals contributing to each cell type. This should be obtainable from the genotyping information.

Reviewer #2 (Recommendations for the authors):

My only 3 issues I would like addressed is that I would prefer Supplementary Figure 2, where they use an HHM to impute the genotypes for each cell, to be adapted to be a main figure along with some text to explain exactly how they do this since it is pretty important for the conclusions. I'd also like some discussion about the ultimate biology that the eQTLS underlie – there are differences in the biology of N2 and CB4956 and it would be nice if they would comment on whether any of this can be explained by eQTLS and the genes affected (even if the answer is negative, totally fine with that). Finally, I note that while eQTLS were previously reported to often show transgressive segregation, this appears not to be the case for the ones found here if I understand correctly and I'd be curious to know the reason– It's a great paper, the writing is clear, the conclusions are strong and the approach is important.

Reviewer #3 (Recommendations for the authors):

This paper is great.

My major recommendation is lifting a bit more of the methods into the main text. Naively when I was reading this I was wondering why the authors had not used a more joint model that could borrow power between cell lines; once I had read into Equations 3 and 4 in eQTL mapping I decided that this was a harder problem than I thought. You might want to hint in the Discussion about taking Equations 3 and 4 into a world where you borrow power between cell types, but it would seem inappropriate to suggest you did this in this paper.

Overall I think you should bring this out in the main text more, and have paragraph on this (maybe Equations 3 and 4 in the main text?).

Specifically I was curious about the X_t_. B_t_ term, which seemed to suggest to me that without this term "bad things happened" (fair enough) but if so I think you should expand more (at least in the supplement) why this term is there.

I am somewhat surprised that the probabilistic genotyping does not go closer to 1 for the homozygous calls (at least in the example shown) and I would.… prefer to see some discussion / exploration of this; surely the homozygous segments must be pretty obvious in the data. If one had some elementary error about homozygous calls from the highly expressed genes…what would happen? Is the issue predominantly about the expression levels to "rule out" het calls? Perhaps the average recombination size and the sparsity of the highly expressed genes make this hard? (And please let us know whether choosing F4 over F2 was in hindsight a good choice.) It feels cute but that there is something here which looks wrong. I don't think this invalidates anything, I am curious about your recommendation for future designs. If you think a future design would be better at the F2 level, I would state that.

In theory you should be able to do ASE. Of course, full ASE will just get mis called (goes homozygous) but partial levels should be "callable" (and goes to the question above – to what extend is a F4 too many switches to model). I would comment on it, even if you don't do it.

---

## [Author Response]

Essential Revisions:1) Clarify and explain the genotyping information/approach that is largely in the supplement. Each reviewer has a comment along these lines.

We have added a section entitled “de novo genotyping of scRNA-seq profiles using a hidden Markov model (HMM)” to the main text. The section describes the HMM genotyping pipeline in more depth. Supplementary Figure 2, showing de novo genotyping results, is now Figure 2 of the main text.

2) Temper some claims in the Introduction and Discussion per the reviewers suggestions.

We’ve modified the text following the reviewers’ comments.

Reviewer #1:

1) We’ve added references to previous studies performing simultaneous genotyping and gene expression measurements.

2) We changed how we present the link between eQTLs and phenotypes in the Introduction.

Reviewer #2:

1) We discuss the biological role of eQTLs and include references to published examples.

Reviewer #3:

1) We discuss possible methodological improvements, including borrowing power between cell types and carrying out allele-specific expression analysis.

We have also revised our manuscript based on the additional comments by the reviewers.

Reviewer #1 (Recommendations for the authors):The concept of using transcriptomes to simultaneously genotype and phenotype samples is actually a somewhat old concept first established in plants in the era of microarrays. I'm not sure if it is appropriate to not cite the original literature establishing this concept. Some acknowledgment of past developments should be provided.

Thank you for this comment. We’ve now added references to a study that used expression microarray data in plants to reconstruct haplotypes (West et al., 2006), a study from our group which similarly used expression microarrays for concurrent genotyping and gene expression measurement in yeast (Ronald et al., 2005), and a study that used expressed variants to demultiplex scRNA-seq profiles (Kang et al., 2018).

Introduction – Is it really valid to say that eQTLs have been proposed to underly genetic associations? Seems like there is a ton of validation studies in Yeast, *C. elegans*, Drosophila, Maize and Arabidopsis that move this beyond the "proposed" stage.

Thank you for this comment. We’ve changed the text accordingly.

“eQTLs have been *found* to underlie genetic associations with complex traits and diseases”

We’ve also added references to a review that discusses several examples for functional roles of eQTLs.

Results – The authors report that the # of eQTLs is correlated with the number of cells of that type in the sequencing data. This suggests that what is actually happening is the # of cells is correlated with the number of independent individuals represented. It would help if the authors could use the genotyping information to estimate the number of individuals contributing to each cell type. This should be obtainable from the genotyping information.

Thank you for this comment. We estimated the number of unique individuals statistically, and our analysis showed that 56%-75% of cells are expected to come from unique individuals. This is due to the large study population (200,000) — nearly four times the number of cells. Notably, that estimate is very conservative since it only looks at overall collision rates in the dataset; collision rates within a cell type should be much lower since they sample a smaller number from the same 200,000 individuals. Consistent with this, we’ve analyzed the genotype correlation matrix of the 1,373 germline cells in our study. We observe that the maximum correlation with another cell is higher than 0.9 for only 39 cells (<3%). Of the 13,219 hypodermis cells (the most abundant cell type), 1,815 (14%) have a maximum correlation of >0.9 with another cell. We now include these estimates for each cell type as a column in Table S1 in Supplementary file 1.

Reviewer #2 (Recommendations for the authors):My only 3 issues I would like addressed is that I would prefer Supplementary Figure 2, where they use an HHM to impute the genotypes for each cell, to be adapted to be a main figure along with some text to explain exactly how they do this since it is pretty important for the conclusions.

Thank you for this suggestion. We’ve added the figure and a section that explains the genotyping to the main text.

I'd also like some discussion about the ultimate biology that the eQTLS underlie – there are differences in the biology of N2 and CB4956 and it would be nice if they would comment on whether any of this can be explained by eQTLS and the genes affected (even if the answer is negative, totally fine with that).

Our results include one possible case in which a biological difference between the strains—body size—could be related to a trans-eQTL hotspot affecting the TGF-β pathway. We’ve added this observation and a discussion of the biological role of eQTLs in phenotypic variation to the Discussion.

Finally, I note that while eQTLS were previously reported to often show transgressive segregation, this appears not to be the case for the ones found here if I understand correctly and I'd be curious to know the reason. It's a great paper, the writing is clear, the conclusions are strong and the approach is important.

Gene expression in recombinant progeny often shows transgressive segregation, where the variance in gene expression across the progeny exceeds that of the parental strains. For example, our previous work in yeast showed that most transcripts show transgressive segregation [Brem and Kruglyak, PNAS 2005]. A likely explanation for these effects is the presence of eQTLs with opposite effects that average out in the parents. We did not compare the variance of gene expression between the segregants and the parents to test for transgressive segregation directly in this dataset, as we’ve done before, due to technical concerns. These concerns include sparseness of the single-cell data as well as confounding factors (total UMI / cell, sequencing coverage, random variation in UMI capture in the library prep, etc.) arising from the fact that we prepared the parental and progeny data in separate experiments. We think studying the cell-type specificity of transgressive effects could be very interesting and hope that future studies will address it.

Reviewer #3 (Recommendations for the authors):This paper is great.My major recommendation is lifting a bit more of the methods into the main text. Naively when I was reading this I was wondering why the authors had not used a more joint model that could borrow power between cell lines; once I had read into Equations 3 and 4 in eQTL mapping I decided that this was a harder problem than I thought. You might want to hint in the Discussion about taking Equations 3 and 4 into a world where you borrow power between cell types, but it would seem inappropriate to suggest you did this in this paper.

We agree that borrowing power between cell types could be beneficial, and we believe this approach needs to be further developed in a future study. As suggested, we’ve added this to the Discussion.

Overall I think you should bring this out in the main text more, and have paragraph on this (maybe Equations 3 and 4 in the main text?).

Thank you for this suggestion. We’ve added language detailing the genotyping. We found, however, that a description of the complex model we use for eQTL mapping did not fit well in the main text, and we feel it is better suited for the Materials and methods section.

Specifically I was curious about the X_t_. B_t_ term, which seemed to suggest to me that without this term "bad things happened" (fair enough) but if so I think you should expand more (at least in the supplement) why this term is there.

We included those terms as we considered the total expression in a cell to be a potential technical source of variation, which we wanted to control for, even though in the examples we looked at, it was not significant.

I am somewhat surprised that the probabilistic genotyping does not go closer to 1 for the homozygous calls (at least in the example shown) and I would.… prefer to see some discussion/exploration of this; surely the homozygous segments must be pretty obvious in the data. If one had some elementary error about homozygous calls from the highly expressed genes…what would happen? Is the issue predominantly about the expression levels to "rule out" het calls?

As the reviewer suggests, the scRNA-seq data’s sparseness means that most variants have very low read counts, making it difficult to rule out het calls. This was part of the reason we’ve opted to use the marginal posteriors directly in the eQTL mapping rather than calling discrete haplotypes.

Perhaps the average recombination size and the sparsity of the highly expressed genes make this hard? (And please let us know whether choosing F4 over F2 was in hindsight a good choice.) It feels cute but that there is something here which looks wrong. I don't think this invalidates anything, I am curious about your recommendation for future designs. If you think a future design would be better at the F2 level, I would state that.

As the reviewer notes, there is a tradeoff when choosing which generation to study between power (which decreases as recombinations accumulate) and resolution (which increases for the same reason). In this study, we prioritized power and studied an F4 population. Unfortunately, studying earlier generations was not feasible since the P0 are manually picked to a plate at the beginning of the experiment to guarantee that all males are from one strain and all females are from the other. We usually obtain 5-fold more progeny at each generation, and often less than that. Therefore, studying 200,000 F2 would require picking at least 8,000 worms manually to start the cross, which would be unfeasible.

In theory you should be able to do ASE. Of course, full ASE will just get mis called (goes homozygous) but partial levels should be "callable" (and goes to the question above – to what extend is a F4 too many switches to model). I would comment on it, even if you don't do it.

Thank you for this suggestion. We agree that ASE could be an interesting application in our framework, and we comment on this possibility in the Discussion.